# Nisin and Organic Acid Salts Improved the Microbial Quality, Extended the Shelf Life, and Maintained the Sensory Attributes of Semidry Beef Luncheon Marketed at Adverse (35–40 °C) Ambient Summer Temperatures

**DOI:** 10.3390/foods12234283

**Published:** 2023-11-27

**Authors:** Ahmed Medhat Elbanna, Rana Fahmi Sabala, Samir Mohammed Abd-Elghany, Kálmán Imre, Adriana Morar, Viorel Herman, Khalid Ibrahim Sallam

**Affiliations:** 1Department of Food Hygiene and Control, Faculty of Veterinary Medicine, Mansoura University, Mansoura 35516, Egypt; drahmedmedhatelbana@gmail.com (A.M.E.); ranafahmi@mans.edu.eg (R.F.S.); drsamir@mans.edu.eg (S.M.A.-E.); 2Department of Animal Production and Veterinary Public Health, Faculty of Veterinary Medicine, University of Life Sciences “King Mihai I” from Timișoara, 300645 Timișoara, Romania; adrianamo2001@yahoo.com; 3Department of Infectious Diseases and Preventive Medicine, Faculty of Veterinary Medicine, University of Life Sciences “King Mihai I” from Timişoara, 300645 Timișoara, Romania; viorel.herman@fmvt.ro

**Keywords:** semidry beef luncheon, nisin, shelf-life extension, microbial quality, sensory attributes, organic acid salts

## Abstract

Semidry beef luncheon may undergo deteriorative changes during storage at ambient temperatures in tropical and subtropical regions including Egypt. This study was conducted in a meat plant in Egypt with the aim of overcoming the economic losses from the returns of spoiled unsold beef luncheon displayed in grocery stores at adverse summer temperatures of 37 °C or more. Ten approaches were applied using different preservatives, comprising sodium nitrite, nisin, potassium sorbate, and organic acid salts (a combination of sodium lactate, sodium acetate, and sodium diacetate). In addition, the product was cooked at different temperatures and was stored for 21 days at 37 °C, during which time the shelf life, microbial quality, pH, and sensory attributes were investigated. By Day 21 of storage, the luncheon contained 50 mg/kg sodium nitrite, 25 mg/kg nisin, and 1000 mg/kg organic acid salts and, when cooked at a final core temperature of 92 °C, exhibited reductions in aerobic plate count, anaerobic plate count, lactic acid bacterial count, and mold and yeast counts by 4.32, 3.54, 3.47, and 1.89 log_10_ CFU/g, respectively, when compared with the control. The sensory attributes and pH were also maintained in the final products of such treatment, with no product return and the avoidance of economic loss. This study presents a novel approach for solving the major problem of the deteriorative changes that occur in semidry luncheon sausage and similar meat products which require rejection with a huge economic loss, especially in tropical and semitropical areas of the world that have similar problems of high climatic temperatures and a low availability of energy or technological resources.

## 1. Introduction

Semidry beef luncheon is a traditional cooked cured meat product in the Mediterranean region in which beef is mixed with variable quantities of carbohydrates such as soy protein, corn starch, spices, and salt, to give the product the unique flavor, color, and consistency accepted by consumers [1]. Traditional semidry beef luncheon, as a popular low-priced cooked meat product, is widely consumed in Egypt. Semidry beef luncheon, however, may undergo deteriorative changes during storage at ambient temperatures in tropical and subtropical regions including Egypt. The shelf life of such products is affected by many factors, the most important of which are the raw materials’ quality, the chemical preservatives added to the product, the temperature used in the manufacturing process, and the storage conditions [2]. Poor storage procedures of processed meat products can be disastrous for food-processing companies and customers. It has been suggested that the spoilage of meat products during storage is one of the most common causes of loss or waste in foods of animal origin in industry [3]. Therefore, the application of good manufacturing practices (GMPs) is essential in choosing raw materials, storage conditions, and distribution, using different preservation approaches to preserve or extend shelf life and ensure the safety and acceptability of the cooked meat products, especially if such products will be eaten without any further cooking.

In Egypt, most cured processed beef products, including semidry beef luncheon, are stored without a refrigerator at room temperature, due to their low water activity and pH value, which inhibit undesirable microbial growth [4]. In tropical and subtropical areas, including Egypt, room temperature can reach 37 °C, which is the optimal temperature for microbial growth. These organisms cause souring, slime formation, and greening of cured beef products; others may produce hydrogen sulfide during growth on vacuum-packaged beef products, causing great spoilage in products with subsequent great economic losses [5]. In addition, the multiplication of foodborne pathogens, for instance, *Listeria monocytogenes*, in semidry beef luncheon, can lead to risks to humans through the consumption of such contaminated foods [6].

Nitrites or nitrates are widely used in the manufacture of most meat products to preserve them from microbial growth. In addition, they produce an effective inhibition for heat-resistant spores, specifically for neurotoxin-producing *Clostridium botulinum* strains, and in cooperation with heat treatment can prevent food spoilage and the multiplication of food-poisoning microorganisms [7]. Nitrates or nitrates have also entered the manufacturing process of the majority of meat products due to their ability to give the acceptable characteristic red color, flavor, and aroma, as well as to retard lipid oxidation by the chelation of metal ions [1,8].

In spite of the wide use of chemical preservatives and synthetic antioxidants for improving the shelf life of meat products because of their low price, stability, and effectiveness, it has been confirmed that nitrates/nitrites are particularly linked to the development of N-nitrosamines, which are carcinogenic and mutagenic nitroso compounds. N-nitrosamines are linked to the development of tumors in different organs, like the liver, stomach, esophagus, and brain, as well as increasing the risk of leukemia, particularly in children [9,10]. The toxicological and carcinogenic aspects resulted from the use of nitrates over the permissible legal limit in the meat industry [11]. Nitrates and nitrites are well documented as being transformed into nitrosamine compounds during cooking, typically in semidry beef luncheon. These nitrosamines can cause a toxic carcinogenic effect on humans. Such toxic effects are of great concern and suggest the introduction of new alternative techniques as substitutes for nitrates and nitrites, or at least to minimize the quantities of them in meat products.

Natural antimicrobials such as chitosan, ε-polylysine, and nisin are considered the new approach to solving the carcinogenic aspects of chemical additives. Nisin is a bacteriocin originating from different *Lactococcus lactis* strains [12]; it is a proteinaceous polypeptide, soluble in aqueous conditions and stable in acidic solutions but unstable in alkali solutions and heat. Thus, nisin is the most commonly used bacteriocin approved by the FDA, with a permissible limit of 25 mg/kg in meat products as an effective preservative [13]. The role of the nisin compound as an alternative to nitrite/nitrate salts in meat products is established in many studies that demonstrate the antimicrobial effects of nisin [14]. Nisin has a potential inhibitory effect against many microorganisms, especially spoilage bacteria and foodborne pathogens like *Listeria monocytogenes* [15,16]. The special effects of nisin can be improved when it is used in combination with other food preservation techniques.

The present study was carried out at a meat plant in Port Said, Egypt, during the regular processing of semidry beef luncheon to overcome the economic losses from the returns of unsold semidry beef luncheon that underwent deteriorative changes during display in grocery stores at summer temperatures of 37 °C or above. The study incorporated different preservative formulations in the luncheon batter, such as sodium nitrite, potassium sorbate, nisin, and organic acid salts, and evaluated the preservatives’ influence on the microbiological quality (aerobic plate counts, anaerobic plate count, lactic acid bacterial count, and mold and yeast count) and sensory attributes of the traditional semidry beef luncheon, which was cooked at various final core temperatures of 87, 90, and 92 °C during 21-days storage at 37 °C in order to identify the most effective formulation, along with the temperature applied to be selected for application during mass production of the product.

## 2. Materials and Methods

### 2.1. Food Preservatives and Their Suppliers

Sodium nitrite, E250, was purchased from the BASF Company, Ludwigshafen, Germany. Nisin, E234, was obtained under the commercial trade name (NovaGARD^®^ NR 100, containing approximately 2.5% *w*/*w* pure nisin) from the Danisco Company, Copenhagen, Denmark. Potassium sorbate, E202, was purchased from the Nantong acetic acid chemical Co., Nantong, China. Organic acid salts (OAS) are a combination of sodium lactate (E325), sodium acetate (E262i), and sodium diacetate (E262ii) produced by Galactic, Brussels, Belgium, and sold under the commercial trade name Galimax Diace-N-47.

### 2.2. Manufacturing of Semidry Luncheon

The raw beef meat cuts used for the manufacturing of traditional semidry beef luncheon were imported (mainly from Brazil) as frozen boneless beef meat cuts which constituted 20 kg/100 kg of the final product. The other ingredients used included water (40 kg/100 kg), vegetable oil (sunflower oil; 6 kg/100 kg), soy protein powder (9.5 kg/100 kg), cornstarch (20 kg/100 kg), sodium chloride (2 kg/100 kg), ascorbic acid (50 g/100 kg), phosphate (300 g/100 kg), garlic (300 g/100 kg), and spice mix (1.85 kg/100 kg). The study was carried out in a local meat plant in Port Said, Egypt. This plant produced about 100 tons of semidry beef luncheon daily. For conducting the present study, five tons of semidry beef luncheon were processed for each of the 10 treatments, and the control (an illustrative diagram for luncheon manufacturing and different treatments applied is presented in the next paragraph).

The frozen beef meat was first ground through a 3 mm-hole grinder plate using a commercial grinder (Wolfking C-250, Grinder, Slagelse, Denmark). Afterward, the ground meat was mixed with regular luncheon ingredients and the selected specific preservative(s) either for the control or the other 10 treatments to be homogenized and chopped together at a temperature of 12 °C using a bowl chopper (Maschinenfabrik, LASKAMakartstraße, Traun, Austria), resulting in the homogeneous beef luncheon emulsion batter. The resulting batters were then stuffed into 95 mm-diameter polyamide casings (Handtmann VF 628, KS Machines-Food Industry Equipment, Berlin, Germany) to yield 3 kg per case, clipped using an automatic double clipper (Polyclip ICA XL, Poly-clip System GmbH & Co. KG, Hattersheim am Main, Germany), and cooked through the modern steam oven to different core temperatures of 87, 90, or 92 °C (Figure 1).

The specific preservatives and final core temperatures (FCT) of cooking for the control and the ten different treatments applied for semidry luncheon were as follows: control (100 mg/kg Sodium nitrite; FCT: 87 °C); T1 (100 mg/kg Sodium nitrite; FCT: 90 °C); T2: (100 mg/kg Sodium nitrite; FCT: 92 °C); T3: (150 mg/kg Sodium nitrite; FCT: 87 °C); T4: (100 mg/kg Sodium nitrite + 12.5 mg/kg Nisin; FCT: 87 °C); T5: (100 mg/kg Sodium nitrite + 25 mg/kg Nisin; FCT: 87 °C); T6: (100 mg/kg Sodium nitrite + 250 mg/kg Potassium sorbate; FCT: 87 °C); T7: (100 mg/kg Sodium nitrite + 500 mg/kg Potassium sorbate; FCT: 87 °C); T8: (100 mg/kg Sodium nitrite + 500 mg/kg organic acid salts; FCT: 87 °C); T9: (100 mg/kg Sodium nitrite + 1000 mg/kg organic acid salts; FCT: 87 °C); T10: (50 mg/kg Sodium nitrite + 25 mg/kg Nisin + 500 mg/kg potassium sorbate + 1000 mg/kg organic acid salts; FCT: 92 °C). All luncheon casings from the differently treated batch samples were subsequently cooled to room temperature through a cold shower.

Five luncheon casings (3 kg each) were taken from each of the ten treatments and the control to be stored for 21 days in an incubator adjusted at 37 °C (a condition imitating the ambient temperature where the product is displayed at the grocery) during which the samples were collected and examined on Days 0, 7, 14, and 21 for the microbiological examination and the sensory evaluation.

The experiment was repeated on three successive days during which five luncheon casings (3 kg each) were taken from each of the ten treatments to carry out the microbiological examination, pH measurement, and sensory evaluation in triplicate.

### 2.3. Microbiological Examination

#### 2.3.1. Sample Preparation for Microbiological Examinations

Beef luncheon samples (five samples from each of the ten treatments) were conveyed to the microbiological laboratory at the plant on Days 0, 7, 14, and 21 of storage. Ten grams from each sample was homogenized with 90 mL of (0.1% *w*/*v*) maximum recovery diluent (MRD, Himedia M1030; HiMedia Laboratories Pvt. Ltd., Mumbai, India) for about two minutes to provide an initial suspension of 1/10 concentration. Serial ten-fold decimal dilutions were then prepared up to 10^−6^.

#### 2.3.2. Aerobic Plate Counts (APCs)

A quantity of 0.1 mL from each dilution was spread onto dried double plates of sterile plate count agar medium (CM0325; Oxoid Ltd., Basingstoke, UK). The inoculated plates were incubated at 30 °C for 72 h and the recovered colonies of aerobic plate counts (APCs) were enumerated in the countable plates and were expressed as log_10_ CFU/g of the tested semidry luncheon samples [17].

#### 2.3.3. Lactic Acid Bacteria Count (LABC)

A quantity of 0.1 mL of previously prepared decimal dilutions was evenly spread onto dried duplicate Petri dishes containing sterile medium of de Man, Rogosa, and Sharpe agar (MRS Agar, Himedia M641; HiMedia Laboratories Pvt. Ltd., Mumbai, India). The inoculated Petri dishes were anaerobically incubated for 48 h at a temperature of 30 °C. The characteristic colonies (pale yellow or whitish colored surrounded by a distinct area) were enumerated and recorded as log_10_ CFU/g of the examined traditional semidry beef luncheon samples in the countable plates [18].

#### 2.3.4. Anaerobic Plate Count (ANPC)

An amount of 0.1 mL from each previously prepared ten-fold decimal dilution was evenly spread on dried duplicated Petri dishes containing a sterile medium of plate count agar (CM0325; Oxoid Ltd., Basingstoke, UK). The inoculated medium was left to dry for 15 min followed by incubation under anaerobic conditions (in an anaerobic jar with gas-producing kits) at 30 °C for 72 h. The colonies of anaerobic plate counts were enumerated and expressed as log_10_ CFU/g of the examined treated semidry luncheon samples [19].

#### 2.3.5. Mold and Yeast Counts (MYC)

An amount of 0.2 mL from each previously prepared ten-fold decimal dilution was spread onto dried duplicate Petri dishes containing sterile Dichloran Rose-Bengal Chloramphenicol Agar (DRBC Agar, Himedia M1881; HiMedia Laboratories Pvt. Ltd., Mumbai, India). The inoculated plates were left to dry for 15 min and incubated at 25 °C for five days. Subsequently, the mold and yeast colonies were enumerated, and the total yeast and mold counts were expressed as log_10_ CFU/g of the examined treated semidry luncheon samples [20].

### 2.4. Measurement of pH

Five grams of semidry beef luncheon samples was added to 20 mL sterile distilled water in a 50 mL beaker, and homogenized for 15 s. The pH value was measured using a formerly calibrated pH meter (Lovibond Senso Direct) with two buffers (pH 7.0 and pH 4.0) with a probe-type electrode (SensoDirect 330 Electrode pH Meter; Thermo Fisher Scientific Inc., Göteborg, Sweden).

### 2.5. Sensory Evaluation

Sensory evaluation of control and treated semidry beef luncheon samples was conducted according to the procedures defined by the American Meat Science Association [21]. The samples were sliced and served in a randomized order to a semi-trained panel of 35 members from the production, quality, and laboratory department in the meat plant. Three sessions were held on each of the four different occasions during the 21-day storage. A nine-point hedonic scoring scale was applied to evaluate the color, odor, taste, texture, and overall acceptability. The panelists were requested to give a numerical value between 1 (indicates extremely unacceptable) and 9 (indicates extremely acceptable). Drinking water was provided between the tasting of each sample to cleanse the palate of the panelists. Sensory evaluation was conducted on Days 0, 7, 14, and 21 of storage.

### 2.6. Statistical Analysis

The microbial counts and the scores of sensory attributes of the control and treated semidry beef luncheon samples were expressed as the mean values of triplicates on the specified examination days during storage. The differences between the means of the microbial categories and pH values among the eleven treatments were determined using generalized linear models (GLMMs) with a Tukey’s multiple comparisons test at the level of 95% significance difference (*p* < 0.05) between treatments and storage time, by using GraphPad PRISM^®^ 9.1.2. (Graph Pad Software Incorporated, San Diego, CA, USA). The sensory evaluation data were analyzed by the two-way GLMM and the multiple comparisons of the means were performed using Tukey’s test to detect the differences among the eleven treatments. In each GLMM, the descriptive variables were the panelists and the sessions (random effects), in addition to the preservatives and processing temperatures (fixed effect). Differences were considered significant at the *p* < 0.05 level.

## 3. Results and Discussion

### 3.1. Effect of Applying Different Preservatives and Cooking to Different Final Core Temperatures (FCT) on the Different Microbial Categories of Semidry Beef Luncheon during Storage at 37 °C

The different microbial categories including APC, ANPC, LABC, and MYC were determined on Days 0, 7, 14, and 21 during the storage of semidry beef luncheon at an adverse temperature of 37 °C.

#### 3.1.1. Aerobic Plate Counts (APCs)

Data concerning the APC in treated and control semidry beef luncheon are presented in Table 1. The mean ± SE values of the initial APC (log_10_ CFU/g) in the semidry beef luncheon on Day 0 of storage ranged from 1.03 in the T2-treated samples to 1.88 in the T8-treated samples. No significant differences were detected in APCs on Day 0 among the different treatments, except between T2- and each of the T8- and T9-treated samples (*p* < 0.01).

On Day 7, significant increases (*p* < 0.01) in the APC were detected for all treatments in comparison with their corresponding initial values, with T10-treated samples exhibiting the lowest count (2.16), followed by T2-treated samples (2.68), while T6- and T4-treated samples revealed the highest counts (3.76 and 3.68, respectively). On Day 14, however, control, T8-, T6-, and T7-treated samples showed the highest APCs (6.23, 5.86, 5.34, and 5.29, respectively), which all exceeded the Egyptian Standards of 5 log_10_ CFU/g set for cooked ready-to-eat meat products [22]. On the other hand, T2-, T3-, T4-, T5-, and T10-treated samples expressed significantly (*p* < 0.01) lower counts in comparison with the other treatments.

The APCs increased as the storage time increased (*p* < 0.01) for both the control and treated samples. By the end of the storage period (Day 21), both the control and treated semidry beef luncheon samples, except the T10-treated samples, exceeded the Egyptian Standards for APC in cooked ready-to-eat meat products. The control samples showed the highest APC of 8.55 log_10_ CFU/g, which was significantly (*p* < 0.01) higher than the corresponding counts in all treated samples. The T10-treated samples achieved a 4.32 log reduction (4.23 vs. 8.55 in control), which seems to be the highest (*p* < 0.01) reduction among the other nine treatments.

The most potent antimicrobial effect, which was expressed by the highest reduction in the APC, was observed in the T10-, T5-, and T4-treated semidry beef luncheon samples and is consistent with the use of nisin, either alone or in combination with organic acid salts. Similar effects concerning the reduction in the APCs were noticed in cooked meat products including cooked ham [23], cooked vacuum-packed emulsion-type sausage [24], cooked beef jerky [25], and processed meat [26] where nisin had been incorporated during these meat products’ processing.

Moreover, a synergistic antimicrobial effect has been reported for nisin in combination with other preservatives with a consequent extension of the shelf life of different food types including frankfurter-type sausages [9], Turkish-style meatballs [27], and raw buffalo meat [28], as well as different meat, chicken, fish, and dairy products [29].

Quite the reverse of our findings, Hampikyan and Ugur [14] reported no significant (*p* > 0.05) difference in APCs in cooked sausages treated with various concentrations of nisin when compared with control samples. Such differences can be referred to the difference in the processing procedures applied.

#### 3.1.2. Anaerobic Plate Counts (ANPC)

The anaerobic plate counts (ANPC) in treated and control semidry beef luncheon are shown in Table 2. The mean ± SE values of the initial ANPC (log_10_ CFU/g) in the semidry beef luncheon on Day 0 of storage ranged from 1.15 in the T5-treated samples to 1.59 in the T8-treated samples. No significant (*p* > 0.05) differences have been detected in ANPCs on Day 0 among the different treatments.

On Day 7, significant increases (*p* < 0.01) in the ANPC were only detected for the control, T6-, and T8-treated samples in comparison with their corresponding initial values. Additionally, control samples exhibited a significantly higher ANPC than that of all the different treatments, while T10-treated samples revealed the lowest ANPC with a 2.54 log difference from the control (1.34 vs. 3.88 log_10_ CFU/g).

By Day 14, however, the control, T7-, T6-, and T8-treated samples showed the highest ANPCs (4.24, 3.28, 3.36, and 3.39 log_10_ CFU/g, respectively), while the T10-treated samples exhibited the lowest count (1.48 log_10_ CFU/g) followed by the T1-, T2-, T9-, T3-, T5-, and T4-treated samples which were all below 3 log_10_ CFU/g.

A similar pattern of growth was detected among the different treatments by the end of the storage period (Day 21), where the control samples showed the highest ANPC of 5.35 log_10_ CFU/g, followed by the T7-, T6-, and T8-treated samples, which all exceeded 4 log_10_ CFU/g. On the other hand, the T10-treated sample expressed the lowest ANPC of 1.81 log_10_ CFU/g, followed by the T3- and T2-treated samples with a count of 2.54 and 2.76 log_10_ CFU/g, respectively. As the storage time increased, a significant (*p* < 0.01) increase was noticed in the ANPCs for both the control and all of the treated semidry beef luncheon samples, except for the T10-treated samples, which showed a log difference of only 0.58 between Day 0 and Day 21 of storage (1.23 vs. 1.81 log_10_ CFU/g), indicating that such a treatment is the best among all of the different treatments applied.

The great reduction (3.54 log_10_ CFU/g) of ANPCs in T10-treated semidry beef luncheon samples obtained on Day 21 of the storage period in comparison with the control is related to the high level of nisin (25 mg/kg) and the organic acid salts (1000 mg/kg) along with cooking to a final core temperature of 92 °C. Comparable reductions in ANPCs were reported in vacuum-packaged pork sausages using hurdle technology stored at ambient temperature (37 ± 1 °C) [30], as well as in cooked beef stored in nisin-containing packaging at 4 °C [31].

Similar to our findings concerning the higher core temperature of cooking (92 °C), the study conducted by Ayadi et al. [32] indicated that cooking turkey meat products (Tunisian salami) samples at higher temperatures produces a reasonable reduction in ANPCs. The lower reduction in ANPCs noticed in the T6-, T7-, and T8-treated semidry beef luncheon samples in addition to the control samples is referred to lacking nisin, along with cooking to a final core temperature of 87 °C.

#### 3.1.3. Lactic Acid Bacterial Count (LABC)

The LABCs in the treated and control semidry beef luncheon are shown in Table 3. The mean ± SE values of the initial LABC (log_10_ CFU/g) in the semidry beef luncheon on Day 0 of storage ranged from 1.12 in each of the T2- and T10-treated samples to 1.62 in the control samples. Interestingly, the initial mean LAB count (1.62 log_10_ CFU/g) in control samples increased progressively to 2.26, 5.24, and 7.35 log_10_ CFU/g on Days 7, 14, and 21 of storage, respectively.

By Day 7, significant increases (*p* < 0.01) in the LABC were detected in T3-, T5-, T6-, and T9-treated semidry beef luncheon samples in comparison with their corresponding counts recorded on Day 0. Interestingly, no significant differences have been detected in the LABC among the different treatments either on Day 0 or Day 7.

By Day 14, however, the T10-treated samples exhibited significantly (*p* < 0.01) lower LAB counts in comparison with the corresponding counts in the other treatments, except those of the T1-, T2-, T5-, and T9-treated samples. By the end of the storage period (Day 21), the LABCs in all of the control and treated semidry beef luncheon samples were significantly (*p* < 0.01) higher than their corresponding counts on Day 14. Nonetheless, the LAB counts in all treated samples were still within the limit of 7 log_10_ CFU/g suggested for cooked sausages by Feng et al. [33], while the control luncheon samples surpassed such limits as it reached 7.35 log_10_ CFU/g.

The gradual significant increase in the LABCs in the control and treated samples during the storage period in the present study was consistent with the results recorded in previous investigations on frankfurter [34] and bologna sausage [35]. The increase in the LABCs is due to the incorporation of carbohydrates in such products, which undergo fermentation in the absence of efficient preservatives creating a suitable condition (acidic pH) for the growth of spoilage bacteria, particularly LAB [5].

The low LAB counts in the T10-, T4-, and T5-treated semidry beef luncheon samples confirmed that nisin was the most potent treatment for reducing LABCs in luncheon samples, with a consequent extension of their shelf life. Several publications revealed the potential reduction effect of nisin on LABCs throughout the storage period of various processed cooked sausages [14], cooked ham [23,36], and vacuum-packed emulsion-type sausage [24], which are consistent with the findings of the present study.

The high LABCs in each of the control, and T1-, T6-, and T7-treated samples were related to the failure of sodium nitrite and potassium sorbate to express an inhibitory effect on such bacteria. This explains that sodium nitrate, when used within its permissible limits, is not sufficient to reduce the LABCs even under proper storage conditions. Several studies reported high LABCs in frankfurter and cooked sausage treated with sodium nitrate and stored at refrigerator temperature [34].

#### 3.1.4. Mold and Yeast Count (MYC)

Mold and yeast counts (MYC; log_10_ CFU/g) in the treated and control semidry beef luncheon during 21-day storage are present in Table 4.

The mean values of the initial MYC in the semidry beef luncheon on Day 0 of storage ranged from 1.13 in the T10-treated samples to 1.56 in the T5-treated samples. By Day 7, the mean MYC ranged from 1.23 in the T10-treated samples to 1.89 in the T5-treated samples. Additionally, no significant differences were found in the MYCs among all samples (including controls and treated ones) tested either on Day 0 or Day 7.

By Day 14, however, significant increases (*p* < 0.01) in the MYC were detected for all treatments and control samples in comparison with their corresponding initial values on Day 0, with the exception of the T10-treated samples, which showed no significant changes between their counts detected during the 21 days of storage.

By the end of the storage period (Day 21), the control, T1, and T2-treated semidry beef luncheon samples exhibited the highest (*p* < 0.01) MYCs among all of the treatments as they recorded 3.75, 3.21, and 3.36 lg_10_ CFU/g, respectively. The other groups of the different treatments were all above 2 lg_10_ CFU/g except the T10-treated samples, which exhibited the lowest count of 1.86 lg_10_ CFU/g indicating that the preservative used along with the FCT of the T10 formulations was the most efficient against mold and yeast.

The T6-, T7-, and T10-treated luncheon samples showed the lowest mold and yeast counts among the different treatments, indicating that potassium sorbate exhibited the most potent effect against mold and yeast growth. The great inhibitory effect of potassium sorbate on the growth of mold and yeast has been previously reported in cooked meat products such as dry-cured ham and Turkish fermented dry-cured sausage [37,38]. In addition, sorbic acid and its salts have broad-spectrum activities against mold and yeast and are safe additives commonly added to meat and poultry products to prolong their shelf life [39].

Similar to the MYCs in the T10-treated semidry beef luncheon samples in the current study, comparable MYCs were reported in frankfurter-type sausage [40,41], as well as in cooked chicken sausages [42], during storage for the same duration, while stored at 4 °C using sodium nitrite as a preservative.

The aforementioned data clarified that using 50 mg/kg sodium nitrite + 25 mg/kg nisin + 500 mg/kg potassium sorbate + 1000 mg/kg organic acid salts in addition to cooking at 92 °C was considered the most potent formula to be applied for the storage conditions in tropical and subtropical areas as in Egypt.

### 3.2. Effect of Applying Different Preservatives and Cooking to Different Final Core Temperatures (FCT) on the pH of Semidry Beef Luncheon during Storage at 37 °C for 21 Days

The pH evaluation of meat products is significant because it affects many sensory attributes such as color, odor, taste, texture, and overall acceptability. The mean ± SE values of the pH (Table 5) in the treated and control semidry beef luncheon indicated no significant differences in the pH among the different groups tested on Day 0, where the pH of all groups was above 6 (between 6.09 in T4-treated samples and 6.22 in T10-treated samples). Likewise, initial pH values higher than 6 were recorded in hot dog sausage [43], as well as in low-fat frankfurters formulated with a healthier lipid combination [44].

On Day 7 in the current study, the control, T1-, T3-, T6-, and T7-treated samples exhibited pH values lower than 6, while the other treatments were still above 6. By Day 14 and thereafter, however, only the T10-treated samples were still above pH 6, while all the other treated and control samples showed pH values lower than 6, with the control samples being the lowest (pH 5.21) at Day 21 of storage followed by the T6-, T7-, and T1-treated samples which exhibited pH values of 5.31, 5.32, and 5.33, respectively.

All of the control and treated samples, except the T10-treated group, revealed a significant (*p* < 0.01) decrease in the pH values (towards acidity), as the storage time increased. The T10-treated luncheon group was almost stable during the whole storage period (Table 5). The change in the pH towards the acidity was expected and coexisted with the higher LABC, with the subsequent production of high amounts of lactic acid especially in the control, T6-, T7-, and T1-treated samples.

Similar to our finding, significant decreases in the pH values (towards acidity) were reported in hot dog sausage stored at 2 °C for 60 days [43], while non-significant decreases in the pH values were recorded in frankfurter-type beef sausage during storage at 4 °C for 14 days [45]. Conversely, significant increases in the pH values (towards alkalinity; from 6.38 to 6.67) were measured in sheep meat sausage during frozen storage for 60 days [46].

### 3.3. Effect of Applying Different Preservatives and Cooking to Different Final Core Temperatures (FCT) on the Sensory Attributes of Semidry Beef Luncheon during Storage at 37 °C for 21 Days

The mean value of the sensory attribute scores in traditional semidry beef luncheon formulated with 11 treatments (10 treatments + control) achieved by the panelists (Table 6) indicated that the use of the different preservatives at different concentrations did not produce any changes in color, odor, taste, texture, and overall acceptability during the first day of storage of the newly processed luncheon. On Day 7, only significantly (*p* < 0.01) higher scores were observed in texture, and overall acceptability in the nisin-containing treatments (T4, T5, and T10) in comparison with the control, while a significant difference was observed in taste for each of T5, T9, and T10 when compared with control samples. On Day 14, however, the color, odor, taste, texture, and overall acceptability scores of all treated luncheon samples exhibited significantly (*p* < 0.01) higher scores in comparison with the controls, with T10-treated samples being the highest. The lowest color, odor, texture, and overall acceptability scores were recorded for the control samples, which were omitted from taste evaluation due to the beginning of deteriorative changes noticed by odor, color, and texture. On Day 21, the color and overall acceptability in all treatments exhibited significantly higher scores in comparison with the control, while the smell attribute showed significantly different scores in all treatments except T1, T6, and T7 when compared with the control. The texture scores were significantly higher in all treatments (except T1) in comparison with the control.

By the end of the storage period (Day 21), the control samples exhibited the lowest sensory scores among all the treatments, while the mean sensory scores showed the lowest values significantly (*p* < 0.01) for all attributes in control, T1-, T3-, T6-, and T7-formulated samples, when compared with those in the other different treatments. Such low scores coexisted with the highest microbial loads and deteriorative changes in the product in odor, texture, and color (Figure 2). The taste evaluation was not carried out in the control, T1-, T3-, T6-, and T7-formulated samples, because of the occurrence of the deteriorative changes. When the deteriorating changes apparently occurred in the control in addition to some treatments (T1, T3, T6, and T7) on Day 21 of storage, the texture of luncheon slices was evaluated by hand to check the degree of hardness and tearing.

At the end of the storage period (Day 21), the sensory scores of the T2-formulated samples (which contained 100 mg/kg sodium nitrite alone and cooked to a final core temperature of 92 °C) were significantly (*p* < 0.01) higher than those observed in the control, T3- (except for the color), T6-, and T7-formulated samples (which were cooked to a final core temperature of 87 °C), indicating that the high core temperature of 92 °C was the most efficient factor irrespective of nitrite or potassium sorbate concentrations added.

The sensory scores of the present study are nearly compatible with those obtained from cooked pork sausage samples on Day 20 of refrigerated storage at 4 °C [47], regardless of the significant variation in storage temperature in their study when compared with that in the current one (4 °C versus 37 °C). Conversely, our sensory findings were incompatible with those obtained by Ozaki et al. [48], who reported higher mean values of sensory attributes in cooked sausage formulated with the same sodium nitrite concentration (150 mg/kg).

On Day 21 of storage, the mean score of the sensory attributes in the T4-formulated samples, which contained 12.5 mg/kg nisin, were higher than those obtained by cooked beef patties samples on Day 21 of chilled storage at 4 °C [49], and also higher than those reported on Day 42 of storage (10 °C) of vacuum-packed emulsion-type sausage treated with 10 mg/kg nisin [24], despite the highly significant differences in the storage temperature (4 or 10 °C versus 37 °C).

The effect of storage time on the sensory attributes of the different treatments indicated that the color, odor, texture, taste, and overall acceptability showed significantly lower scores in the control, T1, T2, T3, T6, T7, T8, and T9 luncheon samples on D14 and thereafter, when compared with the corresponding scores in the first two weeks of storage. On the other hand, T4, T5, and T10 luncheon samples sustained all of the sensory attribute scores over the whole storage period (Table 6).

Our sensory score findings were comparable with those obtained by Fernández-López et al. [34], who reported the mean sensory score of pork cooked-meat products (frankfurter) treated with 150 mg/kg sodium nitrite on Day 21 of refrigerated storage, despite the significant variance in storage temperature and added sodium nitrite concentrations. Likewise, Baldin et al. [50] reported nearly similar sensory scores of mortadella sausage on the last day of the experiment of refrigerated storage (4 °C).

The T10-formulated samples (which contain 50 mg/kg sodium nitrite + 25 mg/kg nisin + 500 mg/kg potassium sorbate + 1000 mg/kg organic acid salts and cooked to a FCT of 92 °C) revealed the most significant (*p* < 0.0001) highest sensory attributes mean scores when compared with those in all different treatments except for the odor and texture scores in the T5-formulated samples (*p* > 0.05).

## 4. Conclusions

The present study concluded that the incorporation of natural antimicrobial agents, such as nisin and organic acid salts, either alone or in a combination, in the processing formulation of traditional semidry beef luncheon resulted in a potent antimicrobial effect against the spoilage microorganisms including aerobic bacteria, anaerobic bacteria, lactic acid bacteria, mold, and yeast. Their incorporation also maintains the sensory attributes of the product throughout the storage times at an adverse storage temperature of 37 °C, with a consequent extension of the product shelf life and avoidance of product returns due to the deteriorative change and the negative consumer perception and hence, economic gains of millions of pounds. The study confirmed that the T10-formulation of semidry beef luncheon which contains nisin (25 mg/kg), organic acid salts (sodium lactate, sodium acetate, and sodium diacetate; 1000 mg/kg), potassium sorbate (500 mg/kg), and a minimal amount of sodium nitrite (50 mg/kg) and cooked to a final core temperature of 92 °C exhibited the most potent effect in retarding the microbial growth, prolonging the shelf life, upholding stability in the product pH, and maintaining the product sensory attributes; and hence it is significantly recommended to be applied for semidry beef luncheon and similar products, especially in tropical and subtropical regions like Egypt and other areas in the world that have similar problems of high temperatures and a low availability of energy or technological resources.

## Figures and Tables

**Figure 1 foods-12-04283-f001:**
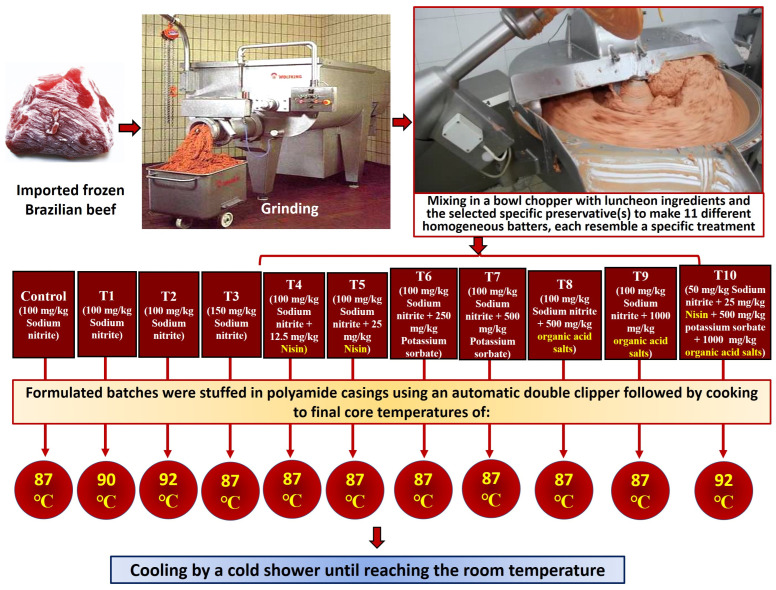
The processing chart of semidry beef luncheon using ten different novel preservative treatments along with various cooking temperatures.

**Figure 2 foods-12-04283-f002:**
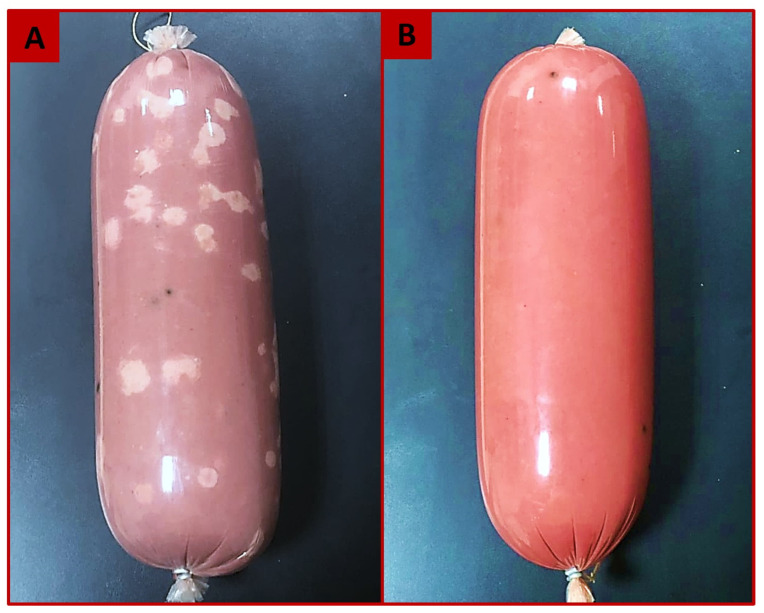
Semidry beef luncheon after 21 storage days at 37 °C. (**A**): Luncheon showing deteriorative changes in appearance, color, and odor. (**B**): Sound luncheon showing the normal appearance, color, and odor.

**Table 1 foods-12-04283-t001:** Effect of applying different preservatives and cooking to different final core temperatures (FCT) on aerobic plate count (APC; log_10_ CFU/g) in semidry beef luncheon during storage at 37 °C for 21 days.

* Treatment	Storage Time (Days)			
	0	7	14	21
Control	1.56 ^ab;w^ ± 0.05	3.25 ^b;x^ ± 0.15	6.23 ^c;y^ ± 0.25	8.55 ^e;z^ ± 0.26
T1	1.15 ^ab;w^ ± 0.18	3.15 ^b;x^ ± 0.16	4.66 ^bc;y^ ± 0.23	6.45 ^bc;z^ ± 0.11
T2	1.03 ^a;w^ ± 0.19	2.68 ^b;x^ ± 0.17	3.65 ^a;y^ ± 0.15	5.33 ^b;z^ ± 0.13
T3	1.55 ^ab;w^ ± 0.15	3.15 ^b;x^ ± 0.18	4.35 ^ab;y^ ± 0.26	6.65 ^c;z^ ± 0.24
T4	1.66 ^ab;x^ ± 0.17	3.68 ^c;y^ ± 0.16	4.13 ^ab;y^ ± 0.19	5.16 ^b;z^ ± 0.13
T5	1.65 ^ab;w^ ± 0.13	3.35 ^b;x^ ± 0.19	4.35 ^ab;y^ ± 0.18	5.03 ^b;z^ ± 0.12
T6	1.34 ^ab;w^ ± 0.17	3.76 ^c;x^ ± 0.07	5.34 ^cd;y^ ± 0.21	7.34 ^cd;z^ ± 0.19
T7	1.69 ^ab;w^ ± 0.16	3.19 ^b;x^ ± 0.18	5.29 ^d;y^ ± 0.16	7.69 ^d;z^ ± 0.09
T8	1.88 ^b;w^ ± 0.12	3.36 ^b;x^ ± 0.12	5.86 ^cd;y^ ± 0.17	6.88 ^c;z^ ± 0.19
T9	1.76 ^b;w^ ± 0.12	3.19 ^b;x^ ± 0.09	4.66 ^b;y^ ± 0.13	5.65 ^b;z^ ± 0.17
T10	1.22 ^ab;x^ ± 0.11	2.16 ^a;y^ ± 0.19	3.66 ^a;z^ ± 0.16	4.23 ^a;z^ ± 0.11

***** Control: (100 mg/kg Sodium nitrite; FCT: 87 °C); T1 (100 mg/kg Sodium nitrite; FCT: 90 °C); T2: (100 mg/kg Sodium nitrite; FCT: 92 °C); T3: (150 mg/kg Sodium nitrite; FCT: 87 °C); T4: (100 mg/kg Sodium nitrite + 12.5 mg/kg Nisin; FCT: 87 °C); T5: (100 mg/kg Sodium nitrite + 25 mg/kg Nisin; FCT: 87 °C); T6: (100 mg/kg Sodium nitrite + 250 mg/kg Potassium sorbate; FCT: 87 °C); T7: (100 mg/kg Sodium nitrite + 500 mg/kg Potassium sorbate; FCT: 87 °C); T8: (100 mg/kg Sodium nitrite + 500 mg/kg organic acid salts; FCT: 87 °C); T9: (100 mg/kg Sodium nitrite + 1000 mg/kg organic acid salts; FCT: 87 °C); T10: (50 mg/kg Sodium nitrite + 25 mg/kg Nisin + 500 mg/kg potassium sorbate + 1000 mg/kg organic acid salts; FCT: 92 °C). Values represent the mean ± SE of triplicate measurements. Mean values with different superscripts from “a” to “e” within the same column are significantly different at *p* < 0.01. Mean values with different superscripts from “w” to “z” within the same row are significantly different at *p* < 0.001.

**Table 2 foods-12-04283-t002:** Effect of applying different preservatives and cooking to different final core temperatures (FCT) on the anaerobic plate count (ANPC; log_10_ CFU/g) in semidry beef luncheon during storage at 37 °C for 21 days.

* Treatment	Storage Time (Days)			
	0	7	14	21
Control	1.52 ^a;x^ ± 0.19	3.88 ^c;y^ ± 0.35	4.24 ^e;y^ ± 0.26	5.35 ^c;z^ ± 0.31
T1	1.45 ^a;x^ ± 0.21	1.86 ^a;xy^ ± 0.11	2.19 ^ab;y^ ± 0.13	3.58 ^b;z^ ± 0.19
T2	1.22 ^a;y^ ± 0.12	1.58 ^a;y^ ± 0.23	2.28 ^ab;z^ ± 0.02	2.76 ^ab;z^ ± 0.15
T3	1.31 ^a;y^ ± 0.19	1.46 ^a;y^ ± 0.16	2.38 ^abc;z^ ± 0.23	2.54 ^ab;z^ ± 0.23
T4	1.51 ^a;x^ ± 0.16	1.88 ^a;x^ ± 0.22	2.58 ^bcd;y^ ± 0.11	3.42 ^ab;z^ ± 0.15
T5	1.15 ^a;x^ ± 0.19	1.65 ^a;x^ ± 0.15	2.57 ^bcd;y^ ± 0.26	3.51 ^b;z^ ± 0.21
T6	1.53 ^a;x^ ± 0.19	2.89 ^b;y^ ± 0.24	3.36 ^de;y^ ± 0.18	4.55 ^bc;z^ ± 0.19
T7	1.34 ^a;w^ ± 0.14	2.12 ^ab;x^ ± 0.14	3.28 ^cd;y^ ± 0.21	4.28 ^bc;z^ ± 0.18
T8	1.59 ^a;x^ ± 0.13	2.85 ^b;y^ ± 0.17	3.39 ^de;y^ ± 0.17	4.88 ^bc;z^ ± 0.12
T9	1.36 ^a;x^ ± 0.17	1.54 ^a;x^ ± 0.31	2.37 ^ab;y^ ± 0.13	3.64 ^b;z^ ± 0.11
T10	1.23 ^a;z^ ± 0.12	1.34 ^a;z^ ± 0.16	1.48 ^a;z^ ± 0.26	1.81 ^a;z^ ± 0.12

***** Control: (100 mg/kg Sodium nitrite; FCT: 87 °C); T1 (100 mg/kg Sodium nitrite; FCT: 90 °C); T2: (100 mg/kg Sodium nitrite; FCT: 92 °C); T3: (150 mg/kg Sodium nitrite; FCT: 87 °C); T4: (100 mg/kg Sodium nitrite + 12.5 mg/kg Nisin; FCT: 87 °C); T5: (100 mg/kg Sodium nitrite + 25 mg/kg Nisin; FCT: 87 °C); T6: (100 mg/kg Sodium nitrite + 250 mg/kg Potassium sorbate; FCT: 87 °C); T7: (100 mg/kg Sodium nitrite + 500 mg/kg Potassium sorbate; FCT: 87 °C); T8: (100 mg/kg Sodium nitrite + 500 mg/kg organic acid salts; FCT: 87 °C); T9: (100 mg/kg Sodium nitrite + 1000 mg/kg organic acid salts; FCT: 87 °C); T10: (50 mg/kg Sodium nitrite + 25 mg/kg Nisin + 500 mg/kg potassium sorbate + 1000 mg/kg organic acid salts; FCT: 92 °C). Values represent the mean ± SE of triplicate measurements. Mean values with different superscripts from “a” to “e” within the same column are significantly different at *p* < 0.01. Mean values with different superscripts from “w” to “z” within the same row are significantly different at *p* < 0.001.

**Table 3 foods-12-04283-t003:** Effect of applying different preservatives and cooking to different final core temperatures (FCT) on lactic acid bacterial count (LAB; log_10_ CFU/g) in semidry beef luncheon during storage at 37 °C for 21 days.

* Treatment	Storage Time (Days)			
	0	7	14	21
Control	1.62 ^a;x^ ± 0.09	2.26 ^a;x^ ± 0.18	5.24 ^c;y^ ± 0.24	7.35 ^e;z^ ± 0.16
T1	1.55 ^a;x^ ± 0.27	2.19 ^a;x^ ± 0.16	3.29 ^ab;y^ ± 0.13	5.28 ^bc;z^ ± 0.19
T2	1.12 ^a;x^ ± 0.36	1.55 ^a;xy^ ± 0.25	2.27 ^a;y^ ± 0.25	4.76 ^b;z^ ± 0.23
T3	1.24 ^a;w^ ± 0.29	2.26 ^a;x^ ± 0.32	3.57 ^b;y^ ± 0.16	5.84 ^c;z^ ± 0.13
T4	1.56 ^a;y^ ± 0.13	2.38 ^a;y^ ± 0.26	3.98 ^b;z^ ± 0.39	4.37 ^ab;z^ ± 0.11
T5	1.35 ^a;w^ ± 0.19	2.35 ^a;x^ ± 0.16	3.27 ^ab;y^ ± 0.16	4.28 ^ab;z^ ± 0.22
T6	1.24 ^a;w^ ± 0.33	2.29 ^a;x^ ± 0.34	4.34 ^bc;y^ ± 0.21	6.97 ^d;z^ ± 0.19
T7	1.61 ^a;x^ ± 0.27	2.39 ^a;x^ ± 0.44	4.28 ^bc;y^ ± 0.26	6.28 ^cd;z^ ± 0.09
T8	1.38 ^a;x^ ± 0.39	2.25 ^a;x^ ± 0.47	4.39 ^bc;y^ ± 0.37	5.88 ^c;z^ ± 0.16
T9	1.36 ^a;x^ ± 0.24	2.54 ^a;y^ ± 0.32	3.37 ^ab;y^ ± 0.23	4.64 ^ab;z^ ± 0.26
T10	1.12 ^a;x^ ± 0.19	1.54 ^a;xy^ ± 0.06	2.28 ^a;y^ ± 0.16	3.88 ^a;z^ ± 0.22

* Control: (100 mg/kg Sodium nitrite; FCT: 87 °C); T1 (100 mg/kg Sodium nitrite; FCT: 90 °C); T2: (100 mg/kg Sodium nitrite; FCT: 92 °C); T3: (150 mg/kg Sodium nitrite; FCT: 87 °C); T4: (100 mg/kg Sodium nitrite + 12.5 mg/kg Nisin; FCT: 87 °C); T5: (100 mg/kg Sodium nitrite + 25 mg/kg Nisin; FCT: 87 °C); T6: (100 mg/kg Sodium nitrite + 250 mg/kg Potassium sorbate; FCT: 87 °C); T7: (100 mg/kg Sodium nitrite + 500 mg/kg Potassium sorbate; FCT: 87 °C); T8: (100 mg/kg Sodium nitrite + 500 mg/kg organic acid salts; FCT: 87 °C); T9: (100 mg/kg Sodium nitrite + 1000 mg/kg organic acid salts; FCT: 87 °C); T10: (50 mg/kg Sodium nitrite + 25 mg/kg Nisin + 500 mg/kg potassium sorbate + 1000 mg/kg organic acid salts; FCT: 92 °C). Values represent the mean ± SE of triplicate measurements. Mean values with different superscripts from “a” to “e” within the same column are significantly different at *p* < 0.01. Mean values with different superscripts from “w” to “z” within the same row are significantly different at *p* < 0.001.

**Table 4 foods-12-04283-t004:** Effect of applying different preservatives and cooking to different final core temperatures (FCT) on mold and yeast counts (log_10_ CFU/g) in semidry beef luncheon during storage at 37 °C for 21 days.

* Treatment	Storage Time (Days)			
	0	7	14	21
Control	1.26 ^a;x^ ± 0.26	1.56 ^a;xy^ ± 0.26	2.32 ^a;y^ ± 0.12	3.75 ^c;z^ ± 0.16
T1	1.23 ^a;y^ ± 0.13	1.63 ^a;y^ ± 0.16	2.56 ^a;z^ ± 0.29	3.21 ^c;z^ ± 0.19
T2	1.26 ^a;x^ ± 0.16	1.53 ^a;x^ ± 0.29	2.55 ^a;y^ ± 0.17	3.36 ^c;z^ ± 0.14
T3	1.24 ^a;y^ ± 0.14	1.86 ^a;yz^ ± 0.12	2.29 ^a;z^ ± 0.18	2.61 ^ab;z^ ± 0.11
T4	1.34 ^a;y^ ± 0.26	1.66 ^a;y^ ± 0.23	2.36 ^a;z^ ± 0.22	2.97 ^b;z^ ± 0.11
T5	1.56 ^a;y^ ± 0.26	1.89 ^a;y^ ± 0.13	2.25 ^a;yz^ ± 0.36	2.87 ^b;z^ ± 0.20
T6	1.29 ^a;x^ ± 0.13	1.55 ^a;xy^ ± 0.12	2.17 ^a;yz^ ± 0.25	2.31 ^a;z^ ± 0.19
T7	1.27 ^a;x^ ± 0.27	1.63 ^a;x^ ± 0.13	2.55 ^a;y^ ± 0.16	2.18 ^a;z^ ± 0.12
T8	1.23 ^a;y^ ± 0.19	1.56 ^a;y^ ± 0.19	2.53 ^a;z^ ± 0.11	2.69 ^b;z^ ± 0.14
T9	1.29 ^a;y^ ± 0.14	1.88 ^a;yz^ ± 0.37	2.46 ^a;z^ ± 0.13	2.71 ^b;z^ ± 0.16
T10	1.13 ^a;z^ ± 0.19	1.23 ^a;z^ ± 0.54	1.56 ^a;z^ ± 0.22	1.86 ^a;z^ ± 0.34

* Control: (100 mg/kg Sodium nitrite; FCT: 87 °C); T1 (100 mg/kg Sodium nitrite; FCT: 90 °C); T2: (100 mg/kg Sodium nitrite; FCT: 92 °C); T3: (150 mg/kg Sodium nitrite; FCT: 87 °C); T4: (100 mg/kg Sodium nitrite + 12.5 mg/kg Nisin; FCT: 87 °C); T5: (100 mg/kg Sodium nitrite + 25 mg/kg Nisin; FCT: 87 °C); T6: (100 mg/kg Sodium nitrite + 250 mg/kg Potassium sorbate; FCT: 87 °C); T7: (100 mg/kg Sodium nitrite + 500 mg/kg Potassium sorbate; FCT: 87 °C); T8: (100 mg/kg Sodium nitrite + 500 mg/kg organic acid salts; FCT: 87 °C); T9: (100 mg/kg Sodium nitrite + 1000 mg/kg organic acid salts; FCT: 87 °C); T10: (50 mg/kg Sodium nitrite + 25 mg/kg Nisin + 500 mg/kg potassium sorbate + 1000 mg/kg organic acid salts; FCT: 92 °C). Values represent the mean ± SE of triplicate measurements. Mean values with different superscripts from “a” to “c” within the same column are significantly different at *p* < 0.01. Mean values with different superscripts from “x” to “z” within the same row are significantly different at *p* < 0.001.

**Table 5 foods-12-04283-t005:** Effect of applying different preservatives and cooking to different final core temperatures (FCT) on pH of semidry beef luncheon during storage at 37 °C for 21 days.

* Treatment	Storage Time (Days)			
	0	7	14	21
Control	6.13 ^a;x^ ± 0.04	5.93 ^b;x^ ± 0.04	5.52 ^c;y^ ± 0.04	5.21 ^e;z^ ± 0.03
T1	6.12 ^a;x^ ± 0.03	5.89 ^b;x^ ± 0.03	5.71 ^bc;y^ ± 0.03	5.33 ^bc;z^ ± 0.03
T2	6.21 ^a;x^ ± 0.04	6.12 ^a;xy^ ± 0.02	5.79 ^ab;y^ ± 0.05	5.61 ^b;z^ ± 0.02
T3	6.18 ^a;w^ ± 0.03	5.91 ^b;x^ ± 0.03	5.61 ^b;y^ ± 0.02	5.42 ^c;z^ ± 0.04
T4	6.09 ^a;y^ ± 0.03	6.05 ^ab;y^ ± 0.04	5.89 ^ab;z^ ± 0.03	5.82 ^ab;z^ ± 0.03
T5	6.11 ^a;w^ ± 0.04	6.10 ^a;x^ ± 0.05	5.91 ^ab;y^ ± 0.03	5.89 ^ab;z^ ± 0.03
T6	6.13 ^a;w^ ± 0.04	5.89 ^b;x^ ± 0.02	5.59 ^bc;y^ ± 0.04	5.31 ^d;z^ ± 0.04
T7	6.21 ^a;x^ ± 0.02	5.91 ^b;x^ ± 0.04	5.58 ^bc;y^ ± 0.03	5.32 ^cd;z^ ± 0.02
T8	6.11 ^a;x^ ± 0.03	6.02 ^ab;x^ ± 0.04	5.71 ^bc;y^ ± 0.04	5.52 ^c;z^ ± 0.03
T9	6.20 ^a;x^ ± 0.04	6.13 ^a;y^ ± 0.03	5.83 ^ab;y^ ± 0.03	5.72 ^ab;z^ ± 0.03
T10	6.22 ^a;x^ ± 0.03	6.18 ^a;xy^ ± 0.03	6.11 ^a;y^ ± 0.02	6.12 ^a;z^ ± 0.04

* Control (100 mg/kg Sodium nitrite; FCT: 87 °C); T1: (100 mg/kg Sodium nitrite; FCT: 90 °C); T2: (100 mg/kg Sodium nitrite; FCT: 92 °C); T3: (150 mg/kg Sodium nitrite; FCT: 87 °C); T4: (100 mg/kg Sodium nitrite + 12.5 mg/kg Nisin; FCT: 87 °C); T5: (100 mg/kg Sodium nitrite + 25 mg/kg Nisin; FCT: 87 °C); T6: (100 mg/kg Sodium nitrite + 250 mg/kg Potassium sorbate; FCT: 87 °C); T7: (100 mg/kg Sodium nitrite + 500 mg/kg Potassium sorbate; FCT: 87 °C); T8: (100 mg/kg Sodium nitrite + 500 mg/kg organic acid salts; FCT: 87 °C); T9: (100 mg/kg Sodium nitrite + 1000 mg/kg organic acid salts; FCT: 87 °C); T10: (50 mg/kg Sodium nitrite + 25 mg/kg Nisin + 500 mg/kg potassium sorbate + 1000 mg/kg organic acid salts; FCT: 92 °C). Values represent the mean ± SE of triplicate measurements. Mean values with different superscripts from “a” to “e” within the same column are significantly different at *p* < 0.01. Mean values with different superscripts from “w” to “z” within the same row are significantly different at *p* < 0.01.

**Table 6 foods-12-04283-t006:** Effect of applying different preservatives and cooking to different final core temperatures (FCT) on the sensory attribute scores of semidry beef luncheon during storage at 37 °C for 21 days.

Storage Day	Sensory Attributes	* Treatments Applied										
		C	T1	T2	T3	T4	T5	T6	T7	T8	T9	T10
Day 0	Color	7.66 ^a^	7.73 ^a^	7.76 ^a^	7.64 ^a^	7.77 ^a^	7.69 ^a^	7.83 ^a^	7.78 ^a^	7.65 ^a^	7.76 ^a^	7.84 ^a^
	Odor	7.74 ^a^	7.57 ^a^	7.45 ^a^	7.29 ^a^	7.55 ^a^	7.50 ^a^	7.65 ^a^	7.79 ^a^	7.61 ^a^	7.46 ^a^	7.57 ^a^
	Taste	7.66 ^a^	7.64 ^a^	7.78 ^a^	7.54 ^a^	7.80 ^a^	7.75 ^a^	7.63 ^a^	7.56 ^a^	7.64 ^a^	7.88 ^a^	7.82 ^a^
	Texture	7.23 ^a^	7.61 ^a^	7.75 ^a^	7.86 ^a^	7.39 ^a^	7.37 ^a^	7.41 ^a^	7.29 ^a^	7.57 ^a^	7.73 ^a^	7.88 ^a^
	Overall acceptability	7.75 ^a^	7.68 ^a^	7.75 ^a^	7.66 ^a^	7.73 ^a^	7.69 ^a^	7.80 ^a^	7.75 ^a^	7.66 ^a^	7.76 ^a^	7.84 ^a^
Day 7	Color	7.16 ^a^	7.23 ^a^	7.46 ^a^	7.14 ^a^	7.67 ^a^	7.59 ^a^	7.23 ^a^	7.28 ^a^	7.25 ^a^	7.56 ^a^	7.77 ^a^
	Odor	7.08 ^a^	7.18 ^a^	7.35 ^a^	7.09 ^a^	7.52 ^a^	7.62 ^a^	6.95 ^a^	6.89 ^a^	7.42 ^a^	7.56 ^a^	7.69 ^a^
	Taste	6.94 ^a^	7.14 ^a^	7.37 ^ab^	7.04 ^a^	7.48 ^ab^	7.71 ^b^	7.03 ^a^	7.16 ^a^	7.46 ^ab^	7.68 ^b^	7.88 ^b^
	Texture	6.68 ^a^	6.81 ^a^	6.95 ^a^	6.49 ^a^	7.53 ^b^	7.69 ^b^	6.54 ^a^	6.47 ^a^	7.07 ^a^	6.94 ^a^	7.92 ^b^
	Overall acceptability	6.75 ^a^	6.68 ^a^	6.75 ^a^	6.66 ^a^	7.63 ^b^	7.79 ^b^	6.80 ^a^	6.75 ^a^	6.86 ^a^	7.02 ^a^	7.95 ^b^
Day 14	Color	3.16 ^a^	4.23 ^b^	5.49 ^b^	4.14 ^b^	6.79 ^c^	7.66 ^d^	4.12 ^b^	4.28 ^b^	5.56 ^b^	6.25 ^b^	7.78 ^d^
	Odor	3.28 ^a^	4.58 ^b^	5.87 ^c^	4.67 ^b^	7.12 ^d^	7.75 ^e^	4.66 ^b^	4.58 ^b^	5.63 ^c^	5.89 ^c^	7.92 ^e^
	Taste	ND	3.88 ^a^	4.88 ^b^	4.28 ^a^	6.85 ^d^	7.45 ^e^	4.18 ^a^	4.35 ^a^	5.57 ^c^	6.78 ^d^	7.75 ^e^
	Texture	2.18 ^a^	3.15 ^b^	4.63 ^d^	3.94 ^c^	7.52 ^f^	7.67 ^f^	4.44 ^cd^	4.12 ^c^	4.85 ^d^	5.71 ^e^	7.81 ^f^
	Overall acceptability	1.87 ^a^	3.45 ^b^	4.44 ^c^	3.58 ^b^	6.55 ^e^	7.53 ^f^	3.88 ^bc^	3.76 ^bc^	5.93 ^d^	5.85 ^d^	8.32 ^j^
Day 21	Color	1.58 ^a^	2.57 ^b^	3.82 ^c^	3.95 ^c^	6.54 ^f^	7.43 ^j^	2.37 ^b^	2.94 ^b^	4.65 ^d^	5.47 ^e^	7.85 ^j^
	Odor	1.52 ^a^	1.74 ^a^	3.87 ^c^	2.38 ^b^	6.72 ^e^	7.65 ^f^	1.66 ^a^	1.72 ^a^	4.46 ^d^	4.85 ^d^	7.92 ^f^
	Taste	ND	ND	3.75 ^a^	ND	5.75 ^c^	6.65 ^d^	ND	ND	4.77 ^b^	5.83 ^c^	7.78 ^e^
	Texture	1.32 ^a^	1.65 ^a^	3.33 ^c^	2.34 ^b^	7.20 ^e^	7.69 ^ef^	2.48 ^b^	2.32 ^b^	3.35 ^c^	4.41 ^d^	7.85 ^f^
	Overall acceptability	1.18 ^a^	1.75 ^b^	3.24 ^d^	2.41 ^c^	5.85 ^g^	7.33 ^h^	2.43 ^c^	1.65 ^b^	4.63 ^e^	5.25 ^f^	8.12 ^i^

* C: control (100 mg/kg Sodium nitrite; FCT: 87 °C); T1 (100 mg/kg Sodium nitrite; FCT: 90 °C); T2: (100 mg/kg Sodium nitrite; FCT: 92 °C); T3: (150 mg/kg Sodium nitrite; FCT: 87 °C); T4: (100 mg/kg Sodium nitrite + 12.5 mg/kg Nisin; FCT: 87 °C); T5: (100 mg/kg Sodium nitrite + 25 mg/kg Nisin; FCT: 87 °C); T6: (100 mg/kg Sodium nitrite + 250 mg/kg Potassium sorbate; FCT: 87 °C); T7: (100 mg/kg Sodium nitrite + 500 mg/kg Potassium sorbate; FCT: 87 °C); T8: (100 mg/kg Sodium nitrite + 500 mg/kg organic acid salts; FCT: 87 °C); T9: (100 mg/kg Sodium nitrite + 1000 mg/kg organic acid salts; FCT: 87 °C); T10: (50 mg/kg Sodium nitrite + 25 mg/kg Nisin + 500 mg/kg potassium sorbate + 1000 mg/kg organic acid salts; FCT: 92 °C). Values represent the mean of the different sensory attribute scores carried out by 35 panelists on three sessions every testing day. Mean values with different superscripts from “a” to “j” within the same row are significantly different (*p* < 0.01). ND: Not detected due to the apparent deteriorating changes.

## Data Availability

Data are contained within the article.

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
