# Peer review of "Nisin and Organic Acid Salts Improved the Microbial Quality, Extended the Shelf Life, and Maintained the Sensory Attributes of Semidry Beef Luncheon Marketed at Adverse (35–40 °C) Ambient Summer Temperatures"

_foods, 2023, doi:10.3390/foods12234283_

Round 1
Reviewer 1 Report
Comments and Suggestions for Authors
Dear Authors,
The manuscript (foods-2700839) submitted for review is quite interesting and I recommend it after revision.
Authors, Please note and address the following comments:
Material and methods
How many samples of semidry beef luncheon were taken for microbiological tests.
I don't know if it should be written like this because sodium nitrite is still approved for use in food industry: „The study lastly recommends the use of nisin and organic acid salts for partial substitution of the excessive concentration of the carcinogenic sodium nitrite that incoporated into cooked cured meat products”.
In my opinion, this sentence should be rewritten.
References
References are cited in accordance with the rules of Foods journal .
I am pleased to recommend this manuscript, but after minor revision.
Reviewer
Author Response
Reviewer #1
The manuscript (foods-2700839) submitted for review is quite interesting and I recommend it after revision.
Dear reviewer,
Thank you so much for your review, kind comments, and valuable suggestions. We have modified the text according to them!
Authors, Please note and address the following comments:
Material and methods
- How many samples of semidry beef luncheon were taken for microbiological tests.
Five samples from each of the ten treatments were taken on days 0, 7, 14, and 21 days of storage. The sample numbers are added in the revised manuscript.
I don't know if it should be written like this because sodium nitrite is still approved for use in food industry: „The study lastly recommends the use of nisin and organic acid salts for partial substitution of the excessive concentration of the carcinogenic sodium nitrite that incorporated into cooked cured meat products”. In my opinion, this sentence should be rewritten.
The sentence is rewritten in the revised manuscript.
References
References are cited in accordance with the rules of Foods journal.
I am pleased to recommend this manuscript, but after minor revision.
Thank you again!
Reviewer 2 Report
Comments and Suggestions for Authors
The work aims to solve a real problem in the industry and I find that very interesting. Furthermore, the work is carried out in Egypt but the results can be applied to many products in other areas of the world that have similar problems of high temperatures and low availability of energy or technological resources. However, the authors have not placed enough emphasis on the novelty or progress of their study. I recommend them to better "sell" the work.
I would have liked to find data regarding oxidation or color changes, I suppose they will be presented in another article.
I have some doubts with the material and methods. The sensory analysis part needs a better description, it is incomplete. It would also be advisable to review the English.
I have included some comments in the text.

Author Response
Reviewer #2
The work aims to solve a real problem in the industry and I find that very interesting. Furthermore, the work is carried out in Egypt, but the results can be applied to many products in other areas of the world that have similar problems of high temperatures and low availability of energy or technological resources. However, the authors have not placed enough emphasis on the novelty or progress of their study. I recommend them to better "sell" the work.
Dear reviewer,
Thank you so much for your review, kind comments, and valuable suggestions. We have modified the text according to them.
A sentence that highlights the importance of our novel approach is added to the abstract as well as in the conclusion of the revised manuscript.
I would have liked to find data regarding oxidation or color changes, I suppose they will be presented in another article.
Exactly, the physical changes including color change as well as the chemical quality including oxidative changes will be presented in another article, which has already been submitted for publication.
I have some doubts about the material and methods. The sensory analysis part needs a better description, it is incomplete. It would also be advisable to review the English.
The sensory analysis is fully described, and the paper will be subjected to English editing via the journal system.
- I have included some comments in the text.
Below, the authors provide answers to the comments of reviewer included in the pdf document during the first revision:
Comment 1: lines 52-54. Solved
Comment 2: lines 79. Solved (further explanation is added).
Comment 3: Line 92. The sentence is moved according to your suggestion.
Comment 4: Line 94. The meaning is clarified in the revised manuscript.
Comment 5: Line 102. The intended achieve is added at the last sentence at the end of the Introduction.
Comment 6: Lines 114-116: Solved and clarified in the revised manuscript.
Lines 114-115: all raised questions are answered and clarified in the revised manuscript.
Line 119. Yes, the plant has a high capacity of producing 100 tons or more of semidry beef luncheon daily.
Line 120: an announcement for the treatment chart is added.
Line 129: 90 and 92 ℃ are newly suggestive temperature while cooking at 87 ℃ was the regular temperature applied in this meat plant.
Line 132-133: Extra information is added in the chart legend.
Lines: 151-153: The sentence is simplified for better understanding.
Line 199: The panel for sensory evaluation was 35 semi-trained persons. Such correction is done in the revised manuscript.
Line 230. Corrected.
Line 231: Tow-way GLM was applied and it is written correctly in the material and methods section.
Line 245. “At” is used instead of “by”.
Line 247: the log number is corrected.
Line 248: “By” is corrected to “At”.
Line 275: The suggested declaration is added.
Line 282: Corrected.
Line 472: The resulting values are already expressed as the mean in the material and methods (Statistical analysis section).
Lines 477- 478. Corrected.
Line 478. Corrected.
Line 479: Corrected.
Lines 482-484, and 492. The texture measurement of luncheon slices was detected by hand to check the degree of hardness and tearing.
Line 497. (Table 6). A paragraph concerning the effect of storage time on the sensory attributes of the different treatments is added in the revised manuscript.
Table 6. ND: means not detected for the taste due to the apparently deteriorating changes. Such a declaration is inserted in the footnote under table 6.
Line 506. The suggested sentence by the reviewer is inserted.
Line 508. Corrected.